# Bloodstream Infections Caused by Extended-Spectrum Beta-Lactamase-Producing *Escherichia coli* in Patients with Liver Cirrhosis

**DOI:** 10.3390/pathogens10010037

**Published:** 2021-01-05

**Authors:** Wen-Chi Chen, Chih-Hsin Hung, Yao-Shen Chen, Jin-Shiung Cheng, Susan Shin-Jung Lee, Fan-Chen Tseng, Ming-Fang Cheng, Jiun-Ling Wang

**Affiliations:** 1Division of Gastroenterology and Hepatology, Department of Internal Medicine, Kaohsiung Veterans General Hospital and School of Medicine, National Yang-Ming University, Taipei 112, Taiwan; wcchen@vghks.gov.tw (W.-C.C.); rcheng@ms2.hinet.net (J.-S.C.); 2Department of Chemical Engineering and Institute of Biotechnology and Chemical Engineering, I-Shou University, Kaohsiung 840, Taiwan; chhung@cloud.isu.edu.tw; 3Division of Infectious Disease, Department of Internal Medicine, Kaohsiung Veterans General Hospital and School of Medicine, National Yang-Ming University, Taipei 112, Taiwan; yschen@vghks.gov.tw (Y.-S.C.); ssjlee@vghks.gov.tw (S.S.-J.L.); 4Graduate Institute of Basic Medical Science, China Medical University and Department of Nursing, National Taipei University of Nursing and Health Sciences, Taipei 112, Taiwan; franc.tseng@gmail.com; 5Department of Pediatrics, Kaohsiung Veterans General Hospital and School of Medicine, Kaohsiung 813, Taiwan; 6School of Medicine, National Yang-Ming University, Taipei 112, Taiwan; 7Department of Nursing, Fooyin University, Kaohsiung 831, Taiwan; 8Department of Internal Medicine, National Cheng Kung University Hospital, Tainan 704, Taiwan; 9Department of Medicine, National Cheng Kung University, Tainan 704, Taiwan

**Keywords:** bacteremia cirrhosis, *Escherichia coli*, extended-spectrum beta-lactamase, sequence type 131

## Abstract

Background: This study aimed to investigate the frequency of sequence type (ST) 131 strains and outcome of cirrhotic patients with bloodstream infections (BSIs) caused by extended-spectrum beta-lactamase-producing *Escherichia*
*coli* (ESBLEC) and non-extended-spectrum beta-lactamase-producing *Escherichia*
*coli* (NESBLEC). Methods: The incidence of ST 131 strains, hospital stay, and 30-day re-admission/mortality were compared between 51 ESBLEC and 51 NESBLEC bacteremic patients with cirrhosis. Results: ST 131 strains were found in 35.3% of the ESBLEC group and 0% of the NESBLEC group (*p* < 0.001). Mean hospital stay was 26.5 days in the ESBLEC group and 17.1 days in the NESBLEC group (*p* = 0.006). Thirty-day re-admission rates were 11.8% in the ESBLEC group and 5.9% in the NESBLEC group (*p* = 0.5). ST 131 strains were associated with 30-day re-admission (odds ratio: 4.5, 95% confidence interval: 1.1–18.9). Thirty-day mortality rate was 31.4% in the ESBLEC group and 23.5% in the NESBLEC group (*p* = 0.4). Conclusion: In patients with cirrhosis, the ESBLEC BSIs group had a higher frequency of ST 131 strains and longer hospital stay than the NESBLEC BSIs group with similar 30-day re-admission/mortality. ST 131 strains were associated with 30-day re-admission.

## 1. Introduction

Patients with liver cirrhosis are susceptible to bacterial infection with the progression of liver dysfunction [1]. Uncontrolled gastrointestinal bleeding, prolonged hospitalization, acute kidney injury, liver failure, and death are among the consequences of bacterial infection in patients with liver cirrhosis [2,3]. Bloodstream infections (BSIs) affect 4% to 21% of the patients with liver cirrhosis, and the 30-day mortality rate can be up to 29% [4,5]. Management of BSIs is an important issue in the care of patients with liver cirrhosis.

The prevalence of multidrug-resistant bacteria can be up to 46% among Gram-negative pathogens cultured from patients with liver cirrhosis [5]. Extended-spectrum beta-lactamase (ESBL)-producing *Escherichia coli* (*E. coli*) (ESBLEC) was the main multidrug-resistant organism identified in patients with liver cirrhosis, which rendered third-generation cephalosporins clinically ineffective and resulted in poor outcomes [6,7]. Recent studies suggested that the increase in resistance to fluoroquinolones and extended-spectrum cephalosporins was linked to the global spread of CTX-M-15-producing sequence type (ST) 131 *E. coli*, which was first described in 2008 [8]. The ST 131 strains accounted for up to two-thirds of multidrug-resistant *E. coli* infection [9]. This highly virulent clonal group was responsible for community-acquired bacteremia as well as health-care associated infections [9]. However, ST 131 *E. coli* infection was rarely evaluated in patients with liver cirrhosis. The objectives of this study were to investigate frequency of ST 131 strains and outcomes in patients with liver cirrhosis and BSIs caused by ESBLEC and NESBLEC.

## 2. Results

### 2.1. Epidemiology

A total of 218 patients with liver cirrhosis and *E. coli* BSIs were identified during the study period (March 2009 to October 2014). Thirteen patients were excluded due to multi-microbial BSIs (2 patients) and isolates collected >72 h after hospitalization (11 patients). Of the 205 patients, 51 patients (24.9%) had ESBLEC BSIs. Another 51 patients with NESBLEC BSIs were randomly selected as the control group. A total of 102 cirrhotic patients were enrolled into this study. The demographic data were similar between both groups, except that more patients in the ESBLEC group had urinary tract infection (UTI) (52.9% vs. 25.5%; *p* = 0.005) and antimicrobial use within 3 months (43.1% vs. 21.6%; *p* = 0.02) than the patients in the NESBLEC group (Table 1).

Antimicrobial susceptibility pattern of these *E. coli* isolates interpreted according to the Clinical and Laboratory Standards Institute guidelines are shown in Table A3 in Appendix A.

### 2.2. ST Distribution of Isolates and blaCTX-M Gene Study

The multiplex polymerase chain reaction assay identified the ST 131 clonal group in 18 (35.3%) patients of the ESBLEC group and 0 (0%) patient of the NESBLEC group (*p* < 0.001) (Table 2). Multi-variate logistic regression with firth correction or with conditional exact analysis were performed to estimate extents of associations between ST 131 clones with ESBLEC and other variables. UTI had a firth correction odds ratio (OR) of 3.3 (*p* = 0.04); ESBLEC had a firth correction OR of 44.2 (*p* = 0.009) with the lower confidence limit of 2.6 and conditional exact analysis OR of 28.8 (*p* < 0.0001) with the lower confidence limit of 5.9. These logistic regression results suggested both UTI and ESBLEC were important factors for being positive for ST 131 clones. ST 73 was found in 0 (0%) patients of the ESBLEC group and 5 (9.8%) patients of the NESBLEC group (*p* = 0.06). ST 95 was identified in 0 (0%) patients of the ESBLEC group and 9 (17.6%) patients of the NESBLEC group (*p* = 0.003). No ST 69 was identified in the ESBLEC group and NESBLEC group.

Of the strains with identified genotype, phylogenetic analysis revealed less incidence of group B2 strain in the ESBLEC group than in the NESBLEC group (21.6% vs. 41.2%; *p* = 0.03) (Table 2).

CTX-M group 9 genes were identified in 25 isolates in the ESBLEC group. CTX-M group 1 genes were identified in 18 isolates in the ESBLEC group (Table 2).

### 2.3. Pulsed-Field Gel Electrophoresis Dendrogram

The dendrogram generated from the Xbal pulsed-field gel electrophoresis profiles was shown in Figure 1 (ESBLEC group) and Figure 2 (NESBLEC group). Using 80% cut-off, there were 33 pulsotypes with 51 ESBLEC isolates. Of the 51 NESBLEC, there were 42 pulsotypes, which showed a great diversity in the pulsed-field gel electrophoresis dendrogram. There were two major clusters (red square) seen in ST 131 strains (black square) (Figure 1). The ST 131 isolates showed more homogeneous pulsotypes than the non-ST 131 isolates.

### 2.4. Outcome of the Patients

The hospital stay of the index BSIs was 26.5 ± 20.7 days in the ESBLEC group and 17.1 ± 12.3 days in the NESBLEC group (*p* = 0.006). In total, inappropriate antimicrobial therapy was administered in 40 (78.4%) of ESBLEC group patients and 8 (15.7%) of NESBLEC group patients (*p* < 0.001). Re-admission due to infection within 30 days was noticed in six patients (11.8%) in the ESBLEC group and three patients (5.9%) in the NESBLEC group (*p* = 0.5). ST 131 clone was the only factor associated with re-admission due to infection within 30 days (OR: 4.5, 95% confidence interval (CI): 1.1–18.9). 

The 30-day mortality rate was 16 patients (31.4%) in the ESBLEC group and 12 patients (23.5%) in the NESBLEC group (*p* = 0.4). The causes of mortality of the patients are shown in Table 3. 

At univariate Cox regression analysis, male gender (OR: 0.4, 95% CI: 0.2–0.9, *p* = 0.04), MELD score (OR: 1.1, 95% CI: 1.1–1.2, *p* < 0.001), spontaneous bacterial peritonitis (OR: 3.3, 95% CI: 1.5–7.3, *p* = 0.003), UTI (OR: 0.2, 95% CI: 0.08–0.6, *p* = 0.005), and inappropriate antimicrobial therapy within 24 h (OR: 2.5, 95% CI: 1.1–5.6, *p* = 0.02) were found to be associated with 30-day mortality. Multi-variate Cox regression analysis found that MELD score (OR: 1.1, 95% CI: 1.1–1.2, *p* < 0.001) and inappropriate antimicrobial therapy within 24 h (OR: 2.5, 95% CI: 1.1–5.5, *p* = 0.03) were significantly associated with 30-day mortality (Table 4). UTI (OR: 0.3, 95% CI: 0.1–0.8, *p* = 0.02) was a protective factor for 30-day mortality.

## 3. Discussion

This study compared the molecular characteristics and outcome between BSIs caused by ESBLEC and NESBLEC in patients with liver cirrhosis. We found that ST 131 clones were more prevalent in the ESBLEC group. MELD score and inappropriate antimicrobial therapy within 24 h were significantly associated with 30-day mortality. ESBLEC was not associated with a higher mortality but was associated with a longer stay in the hospital. These results were able to provide better knowledge about the management of patients with liver cirrhosis and BSIs caused by ESBLEC.

ESBL-producing bacteria was present in 48% of BSIs due to *E. coli* in liver cirrhosis patients [10]. Health care exposure, urinary catheter use, previous antimicrobial therapy, a longer duration of hospital stay, impaired consciousness, histamine-2 blocker use, and immunosuppression were reported to be the risk factors for BSIs due to ESBLEC in different populations [7,11,12]. We found that UTI and use of antimicrobials within 3 months was associated with ESBLEC BSIs in patients with liver cirrhosis, which was similar to the previous studies [12,13,14]. ST 131 dominated among ESBLEC from human blood isolates (64%) in UK [15]. However, ST 131 *E. coli* infection has been rarely investigated in patients with liver cirrhosis, although this virulent strain spreads worldwide. In a cohort of patients with spontaneous bacterial peritonitis [16], ST 131 was found in 2.7% of the cases, which is too few to evaluate the impact of ST 131 clone. ST 131 strains can efficiently colonize gut and persist long term [17], which may lead to invasive disease in cirrhosis. 

Our cohort, focusing on all bacteremia patients with liver cirrhosis, found that UTI was associated with the ST 131 clone. This result corresponds with the capacity of antibiotic resistant ST 131 to produce numerous virulence factors associated with UTI [18]. These findings indicated that ST 131 infection should be considered in patients with liver cirrhosis presenting with UTI caused by ESBLEC. Furthermore, positivity for the virulence genes *iha, hlyD, sat, iutA, fyuA, malX, ompT*, and *traT* was associated with ST 131 [19]. ST 131 clones were associated with a higher risk of re-admission due to infection. The result corresponded to the study by Johnson et al., which showed that ST 131 clones were associated with subsequent hospital admission and new infection [20]. We did not know whether the higher re-admission rate in the patients infected with ST 131 strains was due to the higher antimicrobial resistance or the higher risk of prolonged bacterial colonization [17,21]. Meanwhile, the possible superinfection of ST 131 clones after the use of broad-spectrum antibiotics such as carbapenem could not be excluded and deserves further investigation for carbapenemase-ESBL-producing *Escherichia coli* isolates [22]. A strategy to control the dissemination of these *E. coli* strains with multiple resistance genes is mandatory.

The 30-day mortality rate was similar between the ESBLEC group and NESBLEC group in our study, which was close to the previous reports of BSIs in patients with liver cirrhosis [5,23]. The MELD score and inappropriate antimicrobial therapy within 24 h were associated with the 30-day mortality, while infectious source from urinary tract was negatively associated with 30-day mortality. Patients with advanced liver cirrhosis had increased mortality due to BSIs [24,25]. An *E. coli* bacteremia multicenter study in France showed host factors and the portal of entry exceed bacterial determinants for predicting *E. coli* bacteremia severity [26]. In cirrhotic patients with spontaneous bacterial peritonitis or bacteremia, some experts suggest empirical use of carbapenems in areas with high prevalence of ESBLEC [27]. Delay in adequate antimicrobial use was found to result in an adverse mortality outcome in BSIs caused by ESBLEC [28]. In our study, the longer hospital stays in the ESBLEC group was probably attributable to the fact that more ESBLEC group patients needed adjustment of antimicrobial therapy. In the case of spontaneous bacterial peritonitis, ESBL-producing *E. coli* and *Klebsiella* species resulted in a worse prognosis than non-ESBL-producing species [6]. However, ESBL-producers did not lead to a worse outcome in our cohort. It is noteworthy that urosepsis caused by ESBLEC had a favorable outcome [29]. The survival was possibly determined by the nature of sepsis because ESBLEC was more frequently associated with UTI, and NESBLEC was more frequently associated with intra-abdominal infection. Similarly, primary urinary tract infection was identified as a protective factor in a previous Korean bacteremia cohort [30].

There were limitations in this study. First, selection bias did exist because of the inherent nature of a retrospective study. Second, the Clinical and Laboratory Standards Institute 2009 breakpoints were used to screen the ESBL producers to maintain the consistency throughout the study period. The incidence of ESBL-producing *E. coli* could have been underestimated, because the MIC ranges for cephalosporins to *Enterobacterales* were changed to a great extent in the Clinical and Laboratory Standards Institute 2011 criteria. Finally, this study was conducted in a *bla*_CTX-M-14_ predominant area, and the results should be interpreted cautiously because of the geographic diversity of *E. coli* strains with variable virulence and resistance genes.

## 4. Materials and Methods 

### 4.1. Antimicrobial Susceptibility Testing

Positive blood culture isolates were sub-cultured to a blood agar plate, and the inoculated plates were incubated at 35 °C in 5% CO2 to enable bacterial colonies to develop. After overnight incubation, a standardized inoculum was transferred from the agar medium to perform antimicrobial susceptibility testing with a VITEK 2 automated system 2 (Vitek AMS; bioMérieux Vitek Systems Inc., Hazelwood, MO, USA) using ID-GN and AST-N277 cards (Durham, NC, USA). The antimicrobial susceptibility testing card AST-N277 was used to test the ESBL producers and antimicrobial susceptibility. Intermediate susceptibility was considered as resistance to antimicrobial therapy. The breakpoints of antimicrobial agents and ESBL definition were determined according to the Clinical and Laboratory Standards Institute standards (CLSI M100-S19, 2009) [31].

### 4.2. Detection of ST131 and bla_CTX-M_ Gene Groups

After retrospective chart review for inclusion and exclusion criteria, polymerase chain reaction and sequencing were performed in March 2016 using a T100^TM^ thermal cycler (Bio-Rad, Hercules, CA, USA). A multiplex polymerase chain reaction assay was used to detect phylogenetic group [32] and ST 69, 73, and 95 in addition to ST 131 [33]. These 4 lineages are the most common ST associated with urinary tract infections and bloodstream infections [34]. Phylogenetic group of the isolates was determined by triplex polymerase chain reaction using a combination of two genes (*chuA* and *yjaA*) and an anonymous DNA fragment. The positive and negative control used isolates confirmed by MLST described in our previous work [35]. The primers for PCR amplification and sequencing of the housekeeping genes (*adk*, *fumC*, *gyrB*, *icd*, *mdh*, *purA*, and *recA*) were synthesized from a commercial company (Genomics Biotech Corp. New Taipei City Taipei, Taiwan) according to the primer sequences given on the *E. coli* MLST database website (https://enterobase.warwick.ac.uk/species/ecoli/allele_st_search). Genotypes of CTX-M-type ESBLs, i.e., *bla*_CTX-M_ groups 1, 2, and 9, were identified by a multiplex polymerase chain reaction assay using specific primers as previously described [36,37]. A specific polymerase chain reaction assay was performed to detect the common group 9 variant (CTX-M-14) and group 1 variant (CTX-M-15) [38,39]. The primers used in the genetic study are shown in Table A1 in Appendix A.

### 4.3. XbaI Pulsed-Field Gel Electrophoresis Dendrogram and Phylogenetic Analysis

The clonal relationships between the isolates were determined by pulsed-field gel electrophoresis of XbaI-digested genomic DNA [40]. The GelCompar software package (version 6.0; Applied Maths, Bionumerics) was used to compare the banding patterns of aggregated data. The strains exhibiting 80% similarity in the banding pattern were considered as having similar or identical electrokaryotypes.

### 4.4. Statistical Analysis

The demographic data of the ESBLEC group and the NESBLEC group patients were expressed as frequency or means with standard deviations. Categorical data were compared using chi-square or Fisher’s exact tests when appropriate. Continuous variables with normal distributions were compared using independent Student’s *t* test. Continuous variables without normal distributions were compared using Mann–Whitney U test. Logistic regression analysis was used to examine the risk factor for ST 131. Cox proportional hazard regression model were constructed to analyze the relationship between the variables and mortality. Significance was defined as *p* < 0.05 for all two-tailed tests. All statistical analyses were performed using the SPSS statistical software package, version 17.0 for Windows (SPSS Inc., Chicago, IL, USA).

## 5. Conclusions

In conclusion, in patients with cirrhosis, ESBLEC BSIs group had a higher frequency of ST 131 strains and longer hospital stay than NESBLEC BSIs group with a similar 30-day re-admission and mortality. ST 131 strains were associated with 30-day re-admission. Inadequate antimicrobial therapy and the severity of liver cirrhosis were associated with 30-day mortality. These findings indicated that ESBLEC BSIs should be considered in patients with cirrhosis and UTI or a history of antimicrobials use. Early appropriate antimicrobial therapy should be administered to shorten the hospital stay and reduce the short-term mortality.

## 6. Patients

The study protocol conformed to the ethical guidelines of the 1975 Declaration of Helsinki (6th revision, 2008), as reflected in a priori approval by Institutional Review Board of Kaohsiung Veterans General Hospital (VGHKS14-CT11-01). The need for informed consent was waived off by Institutional Review Board of Kaohsiung Veterans General Hospital. The results of this study were presented in part at the 2016 APASL single topic conference on hepatitis C.

From March 2009 to October 2014, all isolates of *E. coli* cultured from bloodstream of patients with liver cirrhosis were tested for ESBLs at Kaohsiung Veterans General Hospital, a 1700-bed medical center in southern Taiwan that provided both primary and tertiary health care. The inclusion criteria were (1) patients aged ≥18 years; (2) previous diagnosis of liver cirrhosis; (3) a blood culture that grew out ESBLEC or NESBLEC; and (4) ESBLEC or NESBLEC strains isolated from the first time of BSIs. Patients with BSIs collected >72 h after hospitalization and polymicrobial bacteremia were excluded from the study. Each qualified cirrhotic patient with ESBLEC bacteremia was randomly matched with a cirrhotic patient with NESBLEC bacteremia admitted during the same period at a 1:1 ratio. A structured data collection form was used to evaluate the patients, and information on their clinical course was obtained from medical records. Demographic data including age, gender, concomitant diseases, clinical characteristics, laboratory findings, sources of BSIs, antimicrobial susceptibilities, antimicrobial therapy, and 30-day outcomes were collected for each patient.

Liver cirrhosis was diagnosed on the basis of a previous liver biopsy or clinical, biochemical, ultrasonography, computed tomography, or magnetic resonance imaging findings. A model for end-stage liver disease (MELD) score was determined as previously reported [32,41]. A positive blood culture with ESBLEC or NESBLEC was considered to be BSIs caused by ESBLEC or NESBLEC. Spontaneous bacterial peritonitis was defined according to the clinical practice guidelines of EASL [42]. The sources of BSIs were determined following the definition by the Centers for Disease Control and Prevention in 2008 [43]. Inappropriate antimicrobial therapy was defined as lack of sensitive antimicrobial therapy administration within 24 h after the onset of BSIs.

Appropriate specimens including blood, urine, sputum, and ascites were obtained for culture. The attending physicians made all of the decisions regarding antimicrobial therapy before or after the results of the susceptibility tests. The initial antimicrobials were mostly selected empirically. Defervescence within 48 h after the initiation of antimicrobial therapy and a negative blood culture was considered as a response to antimicrobial therapy.

## Figures and Tables

**Figure 1 pathogens-10-00037-f001:**
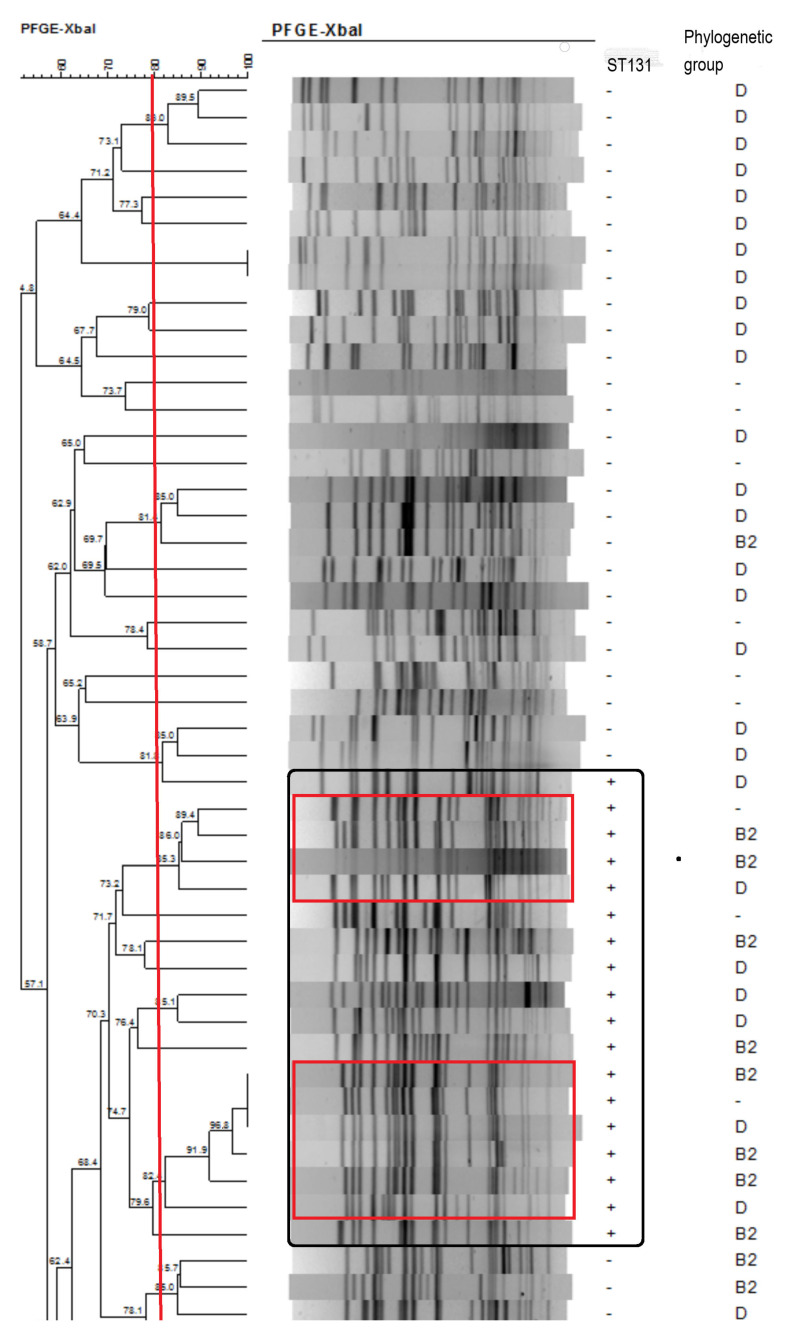
Dendrogram generated from the Xbal pulsed-field gel electrophoresis profiles in ESBL isolates.

**Figure 2 pathogens-10-00037-f002:**
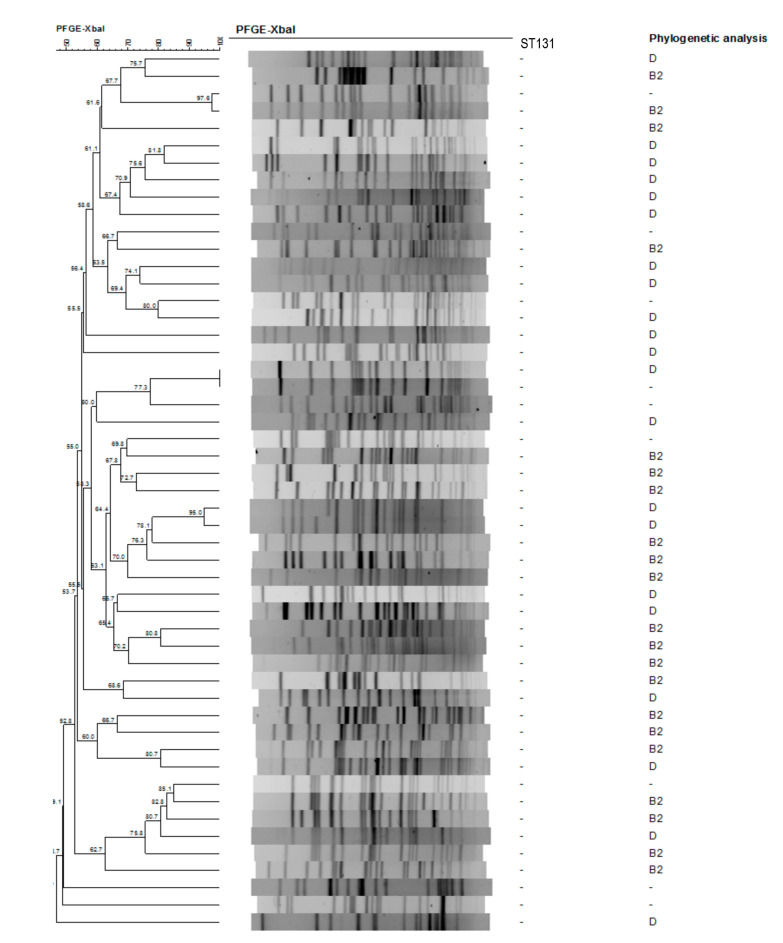
Dendrogram generated from the Xbal pulsed-field gel electrophoresis profiles in non-ESBL isolates.

**Table 1 pathogens-10-00037-t001:** Demographic data of patients with liver cirrhosis and bacteremia caused by extended-spectrum beta-lactamase *Escherichia coli* and non-extended-spectrum beta-lactamase *Escherichia coli*.

Variables	ESBLEC (*n* = 51)	NESBLEC (*n* = 51)	*p* Value
Age (year)	69.1 ± 13.8	67.3 ± 13.7	0.5
Male gender	27 (52.9%)	26 (55.6%)	0.8
Etiology of cirrhosis			
Hepatitis B	13 (25.5%)	16 (17.7%)	0.5
Hepatitis C	23 (45.1%)	23 (45.1%)	1.0
Alcohol	11 (21.6%)	10 (19.6%)	0.8
Others	9 (17.7%)	7 (13.7%)	0.6
Serum albumin (mg/dL)	2.7 ± 0.6	2.7 ± 0.6	0.9
Serum total bilirubin (mg/dL)	3.0 ± 5.2	3.1 ± 3.6	0.9
Prothrombin time-prolonged (sec.)	3.2 ± 3.7	3.4 ± 3.1	0.8
Serum creatinine (mg/dL)	1.9 ± 1.7	1.7 ± 1.6	0.5
Ascites	28 (54.9%)	33 (64.7%)	0.4
Hepatic encephalopathy	8 (15.7%)	9 (17.6%)	0.8
Child–Pugh score	8.2 ± 2.2	8.9 ± 2.5	0.2
MELD ^a^ score	15.7 ± 7.2	15.6 ± 7.0	0.9
Diabetes	18 (35.3%)	26 (51.0%)	0.1
Cardiovascular disease	30 (58.8%)	24 (47.0%)	0.2
Heart failure	4 (7.8%)	3 (5.9%)	1.0
Sources of BSIs ^b^			
Primary bacteremia	12 (23.5%)	16 (31.4%)	0.4
Spontaneous bacterial peritonitis	6 (11.8%)	10 (19.6%)	0.3
Urinary tract infection	27 (52.9%)	13 (25.5%)	0.005
Intraabdominal infection	2 (3.9%)	7 (13.7%)	0.2
Biliary tract infection	3 (5.9%)	4 (7.8%)	1.0
Pneumonia	1 (2.0%)	1 (2.0%)	1.0
Healthcare associated risk			
Prior admission	29 (56.9%)	20 (39.2%)	0.1
Urinary catheterization	5 (9.8%)	3 (5.9%)	0.7
Central venous line	3 (5.9%)	2 (3.9%)	1.0
Endotracheal tube	1 (2.0%)	0 (0%)	1.0
Nasogastric tube	3 (5.9%)	1 (2.0%)	0.6
Endoscopic therapy	2 (3.9%)	3 (5.9%)	1.0
Surgery	2 (3.9%)	1 (2.0%)	1.0
Antibiotics use within 3 months	22 (43.1%)	11 (21.6%)	0.02

^a^ MELD: model for end-stage liver disease. ^b^ BSIs: Bloodstream infections.

**Table 2 pathogens-10-00037-t002:** Genotype of bacteria in patients with liver cirrhosis and bacteremia caused by extended-spectrum beta-lactamase *Escherichia coli* and non-extended-spectrum beta-lactamase *Escherichia coli*.

		ESBLEC (*n* = 51)	NESBLEC (*n* = 51)	*p* Value
**Phylogenetic B2 group ***		11 (21.6%)	21 (41.2%)	0.03
Phylogenetic D group *		29 (56.9%)	21 (41.2%)	0.1
ST 73 ^#^		0 (0%)	5 (9.8%)	0.06
ST 95 ^#^		0 (0%)	9 (17.6%)	0.003
ST 131 ^#^		18 (35.3%)	0 (0%)	<0.001
Pan-CTX-M ^&^		42	NA	NA
CTX-M-group-1	CTX-M-(3, 15)	18	NA	NA
CTX-M-group-2		0	NA	NA
CTX-M-group-8		4	NA	NA
CTX-M-group-9	CTX-M-14	25	NA	NA
Non-CTX-M-group-(1, 2, 8, 9)		10	NA	NA

* Phylogenetic group not detected: ESBL *n* = 11, non ESBL *n* = 9; ^#^ ST not detected: ESBL *n* = 33, non ESBL *n* = 37. ^&^ Some cases had mixed CTX-M group.

**Table 3 pathogens-10-00037-t003:** Outcome of patients with liver cirrhosis and bacteremia caused by extended-spectrum beta-lactamase *Escherichia coli* and non-extended-spectrum beta-lactamase *Escherichia coli*.

	ESBLEC Group (*n* = 51)	NESBLEC Group (*n* = 51)	*p* Value
Index hospital stay (days)	26.5 ± 20.7	17.1 ± 12.3	0.006
Re-admission due to infection within 30 days	6	3	0.5
Mortality within 30 days	16	12	0.4
Hepatoma	1	2	
Liver failure	2	0	
Biliary tract infection	0	1	
Pneumonia	2	1	
Acute respiratory distress syndrome	1	0	
Spontaneous bacterial peritonitis	2	2	
Gastric ulcer bleeding	1	0	
Hepatorenal syndrome	1	2	
Sepsis	4	2	
Variceal bleeding	0	2	
Urinary tract infection	2	0	

**Table 4 pathogens-10-00037-t004:** Multi-variate analysis of the factors associated with 30-day mortality in patients with liver cirrhosis and bacteremia caused by extended-spectrum beta-lactamase *Escherichia coli* and non-extended-spectrum beta-lactamase *Escherichia coli*.

Variables	Odds Ratio	95% Confidence Interval	*p* Value
Male gender	0.8	0.3–1.8	0.5
MELD ^a^ score	1.1	1.1–1.2	<0.001
Urinary tract infection	0.3	0.1–0.8	0.02
Spontaneous bacterial peritonitis	2.0	0.8–4.5	0.1
Inappropriate antimicrobial therapy	2.5	1.1–5.5	0.03

^a^ MELD: model for end-stage liver disease.

## Data Availability

The data presented in this study are contained within the article or Appendix A.

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
