# Peer review of "Bloodstream Infections Caused by Extended-Spectrum Beta-Lactamase-Producing Escherichia coli in Patients with Liver Cirrhosis"

_pathogens, 2021, doi:10.3390/pathogens10010037_

Round 1

Reviewer 1 Report

Article was substantially corrected by Authors. In line 91 I don't understand XXX in the sentence "No ST xxx was identified in ESBL E. coli group 91 and non-ESBL E. coli group". Figures 1 and 2 are unclear and poor quality. In Methods should be describe on which agar, and in which conditions were cultured bacteria before VITEK testing. Which thermocycler (name and producer) was used for PCR? In Supplementary Tables 2 and 3, I suggest write "Resistance rates", instead of "Non-susceptibility rates". In Discussion lacks of the new articles, from recent years 2015-2020. Almost all articles are older than 10 years. In Discussion should be added and cited the following new articles: https://journals.tmkarpinski.com/index.php/ejbr/article/view/259  https://www.thelancet.com/journals/laninf/article/PIIS1473-3099(19)30273-7/fulltext  https://www.sciencedirect.com/science/article/pii/S168411821830433X  https://www.ncbi.nlm.nih.gov/pmc/articles/PMC6254670/  https://www.ncbi.nlm.nih.gov/pmc/articles/PMC4419095/ 

Most of References are old, lack of articles from previous 5 years.

Author Response

  1. Article was substantially corrected by Authors. In line 91 I don't understand XXX in the sentence "No ST xxx was identified in ESBL E. coli group 91 and non-ESBL E. coli group".

Reply: We apologize for the error in the original manuscript. The sentence is re-written as “No ST 69 was identified in ESBL E. coli group and non-ESBL E. coli group”. (line 91)

  1. Figures 1 and 2 are unclear and poor quality.

Reply: We have improved the quality of Figures 1 and 2 as suggested.

  1. In Methods should be describe on which agar, and in which conditions were cultured bacteria before VITEK testing.

Reply: Positive blood culture isolates were subcultured to a blood agar plate and the inoculated plates were incubated at 35°C in 5% CO2 to enable bacterial colonies to develop. After overnight incubation, a standardized inoculum was transferred from the agar medium to perform antimicrobial susceptibility testing with a VITEK 2 automated system 2 (Vitek AMS; bioMérieux Vitek Systems Inc., Hazelwood, MO) using ID-GN and AST-N277 cards (Durham, NC, US). ) (line 198-202)

  1. Which thermocycler (name and producer) was used for PCR?

Reply: The polymerase chain reaction and sequencing were performed in March 2016 using a T100 Thermal Cycler (Bio-Rad, USA) (line 208-9)

  1. In Supplementary Tables 2 and 3, I suggest write "Resistance rates", instead of "Non-susceptibility rates

Reply: "Non-susceptibility rates" are placed with "Resistance rates" in Supplementary Tables 2 and 3 as the reviewer suggested. (line 286& 290)

  1. In Discussion lacks of the new articles, from recent years 2015-2020. Almost all articles are older than 10 years. In Discussion should be added and cited the following new articles: https://journals.tmkarpinski.com/index.php/ejbr/article/view/259https://www.thelancet.com/journals/laninf/article/PIIS1473-3099(19)30273-7/fulltext  https://www.sciencedirect.com/science/article/pii/S168411821830433X  https://www.ncbi.nlm.nih.gov/pmc/articles/PMC6254670/  https://www.ncbi.nlm.nih.gov/pmc/articles/PMC4419095/ 

Most of References are old, lack of articles from previous 5 years.

Reply: We have added the new references in the Discussion section as the reviewer suggested. The order of the References is properly changed.

(line 150,160,168,176)

Reviewer 2 Report

Nothing to add

Author Response

Reply: We thank the Comments and Suggestions from the reviewer.

Reviewer 3 Report

The emergence of antimicrobial resistance among E. coli that cause BSIs is a serious problem difficult to treat. Especially the production of extended-spectrum β-lactamases by E. coli isolates has become a clinical concern worldwide. The Authors investigated frequency of ST 131 strains and outcomes in the patients with liver cirrhosis and BSIs caused by ESBL E. coli and non-ESBL E. coli.

Major comments

The sections Introduction and Materials and methods should be completed with the important information as indicated below.

Specific comments

Introduction

There is no information about the importance of O25b serotype in this study.

Why did the Authors analyze the ST 69, 73 and 95?

Results

2.2. ST distribution of isolates and blaCTX-M gene study

Line 91

No ST xxx was identified? Probably should be: No ST 69 was identified.

2.3. Pulsed-field gel electrophoresis dendrogram

Line 106

“Ten isolates within the two major pulsotype” - the term “isolates” is confusing here, because there is 18 isolates with ST131 type. Rather the term pulsotypes within clusters/groups would be better.

It seems also that using 80% cut-off, there is seven pulsotypes within two major clusters (n=6 and n=1, respectively).

Discussion

Line 140

Should be: between BSIs caused by ESBL E. coli and non-ESBL E. coli

Materials and Methods

There is no information about E. coli strains used in this study in this section.

In the Patients section there is information that from March 2009 to October 2014, all isolates of E. coli cultured from bloodstream of patients with liver cirrhosis were tested  for  ESBLs, but it is not enough.

In the Materials and methods section should be information when the PCR reaction and sequencing were performed.

4.2. Detection of ST131 and blaCTX-M gene groups

There is no information in the manuscript why the Authors screened E. coli strains for the O25b serotype and there are no results of this screen.

There is no information about the controls used in PCR reactions.

There is no information about sequencing to determine the sequence types.

4.3. XbaI pulsed-field gel electrophoresis dendrogram and phylogenetic analysis

There is no literature reference to the conditions of pulsed-field gel electrophoresis.

Author Response

Major comments

The sections Introduction and Materials and methods should be completed with the important information as indicated below.

 Reply: thanks a lot for your suggestions. We have revised as below.

Specific comments

Introduction

There is no information about the importance of O25b serotype in this study.

Reply: Due to lack of new information in this study,we deleted O25b description.

Why did the Authors analyze the ST 69, 73 and 95?

Reply: Because these ST types were most common in human bacteremia as previous study. These 4 lineages are the most common ST associated with urinary tract infections and bloodstream infections[Ref 34] ( line 214)

Results

2.2. ST distribution of isolates and blaCTX-M gene study

Line 91. No ST xxx was identified? Probably should be: No ST 69 was identified.

Reply: The sentence is re-written as “No ST 69 was identified in ESBL E. coli group and non-ESBL E. coli group”.

 2.3. Pulsed-field gel electrophoresis dendrogram

Line 106

“Ten isolates within the two major pulsotype” - the term “isolates” is confusing here, because there is 18 isolates with ST131 type. Rather the term pulsotypes within clusters/groups would be better.

It seems also that using 80% cut-off, there is seven pulsotypes within two major clusters (n=6 and n=1, respectively).

Reply: We have revised. There were two major clusters (red square) seen in ST 131 strains (black square) (line 107-8)

Discussion

Line 140

Should be: between BSIs caused by ESBL E. coli and non-ESBL E. coli

Reply: We feel sorry for the error. The sentence is re-written as “This study compared the molecular characteristics and outcome between BSIs caused by ESBL E. coli and non-ESBL E. coli in patients with liver cirrhosis.”

Materials and Methods

There is no information about E. coli strains used in this study in this section.

Reply:

In the Patients section there is information that from March 2009 to October 2014, all isolates of E. coli cultured from bloodstream of patients with liver cirrhosis were tested for ESBLs. Also see line 202-206.

Reply:

In the Materials and methods section should be information when the PCR reaction and sequencing were performed.

Reply: The PCR reaction and sequencing were performed in March 2016. This is added to the session “4.2. Detection of ST131 and blaCTX-M gene groups”

4.2. Detection of ST131 and blaCTX-M gene groups

There is no information in the manuscript why the Authors screened E. coli strains for the O25b serotype and there are no results of this screen.

Reply: The O25 is the most common O type in ST131. We performed O25 PCR in ST131 to confirm the epidemic strains. However, the result did not add new information, so we decided to delete this part in this work.

There is no information about the controls used in PCR reactions.

Reply: We have added the sentence. The positive and negative control were using isolates confirmed by MLST described in our previous work [Ref 35] (line 217-8).

There is no information about sequencing to determine the sequence types.

Reply: We used multiplex PCR to determine the common four type and the positive/negative control isolates were from MLST result.(line 217-8)

4.3. XbaI pulsed-field gel electrophoresis dendrogram and phylogenetic analysis

There is no literature reference to the conditions of pulsed-field gel electrophoresis.

Reply: We have added the reference. (now in Ref 40)

Reviewer 4 Report

In this manuscript, the authors compared and contrasted BSIs caused by ESBL-producing vs. non-ESBL producing E. coli in patients with liver cirrhosis. They determined the incidence of different STs and found ST131 was associated with the ESBL-producers only.  Those patients with EBSL-producer infection had longer hospital stays, were more likely to get readmitted, which was associated with ST131, and had a higher 30 day mortality rate.   Those with ESBL-producer infections were more likely to receive in appropriate therapy.  The manuscript presents data that will allow clinician to better manage their patients with liver cirrhosis and hopefully initiate appropriate therapy sooner if ST131 is causing UTI in these patients.  

Comments:

1.  The authors need to include and discuss the following references:

Oncotarget . 2017 Dec 13;9(87):35780-35789. doi:0.18632/oncotarget.23200. eCollection 2018 Nov 6.  Bloodstream infection due to Escherichia coli in liver cirrhosis patients: clinical features and outcomes.

EBioMedicine. 2018 Sep;35:76-86. doi: 10.1016/j.ebiom.2018.08.029. Epub 2018 Aug 20. Impact of host-pathogen-treatment tripartite components on early mortality of patients with Escherichia coli  bloodstream infection: Prospective observational study

2.  Line 31, for accuracy, authors should include ESBL-"producing" E. coli and non-ESBL-"producing" E. coli or similar term to "producing"

3.Line 52, Please re-write for English, not correct as written.

4. Line 67, please include dates for study period here.

5. Line 91,  should this be "ST69", what is "ST xxx"

6. Line 187, please refer to "Enterobacteriaceae" as "Enterobacterales"

7. Lines 188 and 206, Please italicize gene names; also for bla genes, "bla" should be italicized and the gene name should be regular font, but subscript.

8.  Line 210, please refer to supplemental table 1, not appendix.  

Author Response

Comments:

  1. The authors need to include and discuss the following references:

Oncotarget . 2017 Dec 13;9(87):35780-35789. doi:0.18632/oncotarget.23200. eCollection 2018 Nov 6.  Bloodstream infection due to Escherichia coli in liver cirrhosis patients: clinical features and outcomes.

EBioMedicine. 2018 Sep;35:76-86. doi: 10.1016/j.ebiom.2018.08.029. Epub 2018 Aug 20. Impact of host-pathogen-treatment tripartite components on early mortality of patients with Escherichia coli bloodstream infection: Prospective observational study.

Reply: The novel references suggested by the reviewer are included and discussed in the Discussion section. (New Ref 10 and 30) (line 148,189)

  1. Line 31, for accuracy, authors should include ESBL-"producing" E. coli and non-ESBL-"producing" E. coli or similar term to "producing"

Reply: We have included ESBL-"producing" E. coli and non-ESBL-"producing" E. coli in line 31 as the reviewer suggested.

3.Line 52, Please re-write for English, not correct as written.

Reply: The text in line 52 is re-written for English as the reviewer suggested. (now in line 53)

  1. Line 67, please include dates for study period here.

Reply: The dates for study period are provided in line 67 as suggested.(now line 68)

  1. Line 91, should this be "ST69", what is "ST xxx"

Reply: We apologize for the error in the original manuscript. The sentence is re-written as “No ST 69 was identified in ESBL E. coli group and non-ESBL E. coli group”.

  1. Line 187, please refer to "Enterobacteriaceae" as "Enterobacterales"

Reply: We have referred "Enterobacteriaceae" as "Enterobacterales" as the reviewer suggested. (Now in line 196)

  1. Lines 188 and 206, Please italicize gene names; also for bla genes, "bla" should be italicized and the gene name should be regular font, but subscript.

Reply: The gene names in Lines 188 and 206, as well as in lines 81 and 200, are corrected as the suggestion by the reviewer. (now line 197 &218)

  1. Line 210, please refer to supplemental table 1, not appendix.  

Reply: We refer to Supplementary Table 1 instead of appendix as the reviewer suggested. (line 221)

Round 2

Reviewer 1 Report

Authors corrected manuscript according to Reviewer's suggetsions.

Author Response

Thank you for your comments.

Reviewer 3 Report

The Authors have revised the paper according to reviewer’s comments. The paper has been improved and requires only minor correction.

Specific comments

Materials and Methods

It is still unclear under what conditions E. coli isolates were stored from 2014 to further analysis (PCR and MLST) performed in 2016.

There is no information where the PCR products were sequenced (what service) and what software was used to compare the sequences.

Author Response

Thank you for your comments.

After retrospective chart review for inclusion and exclusion criteria, the polymerase chain reaction and sequencing were performed in March 2016 using a T100TM Thermal Cycler (Bio-Rad, USA).

The primers for PCR amplification and sequencing of the housekeeping genes (adk, fumC, gyrB, icd, mdh, purA, and recA) were synthesized from a commercial company (Genomics Biotech Corp. New Taipei City Taipei, Taiwan) according to the primer sequences given on the E. coli MLST database website. https://enterobase.warwick.ac.uk/species/ecoli/allele_st_search.

We have added these sentences on Line 209 and line 216-220.